# In Newborn Infants a New Intubation Method May Reduce the Number of Intubation Attempts: A Randomized Pilot Study

**DOI:** 10.3390/children8070553

**Published:** 2021-06-26

**Authors:** Marlies Bruckner, Nicholas M. Morris, Gerhard Pichler, Christina H. Wolfsberger, Stefan Heschl, Lukas P. Mileder, Bernhard Schwaberger, Georg M. Schmölzer, Berndt Urlesberger

**Affiliations:** 1Division of Neonatology, Department of Pediatrics and Adolescent Medicine, Medical University of Graz, Auenbruggerplatz 34/2, 8036 Graz, Austria; nicholas.morris@medunigraz.at (N.M.M.); gerhard.pichler@medunigraz.at (G.P.); christina.wolfsberger@medunigraz.at (C.H.W.); lukas.mileder@medunigraz.at (L.P.M.); bernhard.schwaberger@medunigraz.at (B.S.); berndt.urlesberger@medunigraz.at (B.U.); 2Research Unit for Neonatal Macro- and Microcirculation, Medical University of Graz, Auenbruggerplatz 34/2, 8036 Graz, Austria; 3Department of Anesthesiology and Intensive Care Medicine, Pediatric Anesthesia, Medical University of Graz, Auenbruggerplatz 34/2, 8036 Graz, Austria; stefan.heschl@medunigraz.at; 4Centre for the Studies of Asphyxia and Resuscitation, Neonatal Research Unit, Royal Alexandra Hospital, 10240 Kingsway Avenue NW, Edmonton, AB T5H 3V9, Canada; georg.schmoelzer@me.com; 5Department of Pediatrics, Faculty of Medicine and Dentistry, University of Alberta, 11405-87 Avenue, Edmonton, AB T6G 1C9, Canada; 6Research Unit for Cerebral Development and Oximetry Research, Medical University of Graz, Auenbruggerplatz 34/2, 8036 Graz, Austria

**Keywords:** intubation, endotracheal tube, ventilation, acute respiratory failure, desaturation, neonatal intensive care unit, neonates

## Abstract

Severe desaturation or bradycardia often occur during neonatal endotracheal intubation. Using continuous gas flow through the endotracheal tube might reduce the incidence of these events. We hypothesized that continuous gas flow through the endotracheal tube during nasotracheal intubation compared to standard nasotracheal intubation will reduce the number of intubation attempts in newborn infants. In a randomized controlled pilot study, neonates were either intubated with continuous gas flow through the endotracheal tube during intubation (intervention group) or no gas flow during intubation (control group). Recruitment was stopped early due to financial and organizational issues. A total of 16 infants and 39 intubation attempts were analyzed. The median (interquartile range) number of intubation attempts and number of abandoned intubations due to desaturation and/or bradycardia were 1 (1–2) and 4 (2–5), (*p* = 0.056) and *n* = 3 versus *n* = 20, (*p* = 0.060) in the intervention group and control group, respectively. Continuous gas flow through the endotracheal tube during intubation seems to be favorable and there are no major unexpected adverse consequences of attempting this methodology.

## 1. Introduction

The current intubation standard procedure includes sedation of the newborn infant, potentially leading to airway instability and diminishing breathing efforts. This causes a discontinuation of air/oxygen flow to the lungs before correct endotracheal tube (ETT) placement and might result in oxygen desaturation and/or bradycardia. During endotracheal intubation severe desaturation occurs in up to 51% of infants [1]. Furthermore, more than two intubation attempts are associated with an increased incidence of severe complications [2,3]. Providing continuous gas flow via the ETT itself during the intubation attempt might improve newborn infants’ stability and thereby increase successful intubation rates.

We hypothesized that newborn infants requiring intubation who receive continuous gas flow during intubation (intervention group) will require less intubation attempts compared to the standard approach without gas flow (control group).

## 2. Materials and Methods

This randomized controlled pilot trial was carried out at the Division of Neonatology, Medical University of Graz, Austria, between October 2016 and October 2020 and registered at clinicaltrials.gov (identifier: NCT04089540). The Regional Committee on Biomedical Research Ethics of the Medical University of Graz approved the study protocol (EC number: 25–282ex12/13).

### 2.1. Study Poplulation

Term and preterm neonates admitted to neonatal intensive care units requiring intubation due to respiratory failure were eligible. Written parental informed consent was obtained prior to inclusion. Neonates with severe congenital malformations of the upper airway and hemodynamically significant congenital cardiovascular malformations were excluded.

### 2.2. Randomization and Blinding

Neonates were randomly assigned 1:1 to an intervention or control group by a computer-generated randomization software (www.randomizer.at), using a block randomization with a block size of 6. Blinding was not possible, considering the type of intervention.

### 2.3. Sample Size

A sample size of 40 infants was arbitrarily designated and authorized by the local ethics committee. Sample size calculations were not performed since no data from previous studies were available.

### 2.4. Interventions

The nasopharyngeal route for intubation was used according to the standard procedure. Infants routinely received 1 mg/kg Propofol (Fresenius Kabi, Bad Homburg vor der Höhe, Germany) intravenously for sedation shortly before intubation. Propofol application could be repeated if needed. In the intervention group, the Neopuff Infant T-Piece Resuscitator (Perivent, Fisher& Paykel Healthcare; New Zealand) was connected to the ETT with the default settings of positive-end expiratory pressure of 5 cmH_2_O and gas flow of 6 L/min ≤ 1000 g, 7 L/min between 1000 and 2000 g, or 8 L/min > 2000 g birth weight. The fraction of inspired oxygen was adjusted for each patient during non-invasive mask ventilation prior to intubation, aiming for a target peripheral arterial oxygen saturation (SpO_2_) of >89%. Continuous gas flow was provided through the ETT by an assisting staff member from insertion of the ETT into the nose until the ETT passed through the vocal cords. Once the ETT was placed in the trachea, continuous gas flow was discontinued and the ventilator was connected to start positive pressure ventilation (PPV). In the control group, no continuous gas flow was provided during intubation. Auscultation and/or exhaled carbon dioxide detection was used to assess correct ETT position. The duration of each intubation attempt was defined as the time from the removal of the face mask until the confirmation of correct ETT placement and was measured by a study team member using a stopwatch.

### 2.5. Intubation Attempt Abortion Criteria

Intubation attempts were stopped if SpO_2_ was <80% and/or the heart rate was <100 beats/min for >5 s. In the intervention group, the ETT was kept inserted in the nostril while the other nostril and the mouth were held closed, and PPV was provided via the ETT. In the control group the ETT was removed and non-invasive mask ventilation was performed until the neonate was stabilized.

### 2.6. Data Collection and Statistical Analysis

Demographics of study patients, intubation characteristics and the parameters of hospital stay were recorded. The primary outcome was the number of intubation attempts. The data are presented as mean (SD) for normally distributed continuous variables and median (IQR) when the distribution was skewed. We used intention-to-treat analysis and compared data using the Student’s t-test for parametric and Mann-Whitney U test for non-parametric comparisons of continuous variables, and the Fisher’s exact test for categorical variables. Statistical analyses were performed with IBM-SPSS-Statistics 24 Software (PSS Inc., Chicago, IL, USA.).

## 3. Results

Patient enrollment and allocation are demonstrated in the flow chart (Figure 1). Sixteen neonates were included; recruitment was stopped early due to financial and organizational issues (loss of equipoise within the recruiting team). Hence, we are reporting the data as a posteriori pilot study. Demographics, intubation parameters, and characteristics of hospital stay are presented in Table 1. Respiratory diagnoses of the included infants were respiratory distress syndrome (*n* = 13), meconium aspiration syndrome (*n* = 1) and pneumothorax (*n* = 2).

### 3.1. Intubation Attempts

A total of 39 intubation attempts (intervention group *n* = 10, control group *n* = 29) were performed. The median (IQR) number of intubation attempts was 1 (1–2) and 4 (2–5) in the intervention and control group, respectively (*p* = 0.056) (Figure 2). The number of abandoned intubation attempts due to desaturation and/or bradycardia was n=3 in the intervention group vs. *n* =20 in the control group (*p* = 0.060).

### 3.2. Secondary Outcomes

The first-pass success rate was 57% vs. 22% (*p* = 0.303) and the Propofol dose median (IQR) 1 (1–2) vs. 2 (2–3) mg/kg (*p* = 0.072) in the intervention compared to the control group, respectively. Duration until successful intubation was 204 (138–300) and 858 (330–924) sec in the intervention and control group, respectively (*p* = 0.113) (Table 1, Figure 2). There were no adverse effects in either groups.

## 4. Discussion

To the best of our knowledge, this is the first randomized controlled trial comparing continuous gas flow through the ETT during intubation with the standard procedure aiming to reduce intubation attempts in newborn infants. This study demonstrated that the concept of continuous gas flow through the ETT during intubation, and the initial results, seem to be favorable and that there are no major unexpected adverse consequences of attempting this methodology. Further on, continuous gas flow through the ETT might result in a reduction of the total number of intubation attempts, fewer abandoned intubation attempts, higher first-pass success rates and less Propofol administration. From a clinical perspective, the study team preferred the intervention intubation method and loss of equipoise was the consequence.

In pediatric patients, continuous oxygen flow through a nasal cannula results in longer periods of normoxemia compared to no oxygen flow during intubations in the operating room [4]. While we did not measure the time to desaturation, we suspect a shorter time until successful intubation and fewer abandoned intubation attempts due to desaturation and/or bradycardia in the intervention group, indicating improved cardiorespiratory stability during intubation. In our center infants routinely receive Propofol intravenously for sedation before intubation. Our standard operating procedure is based on a randomized controlled trial demonstrating no differences in heart rate when using the Propofol compared to the morphine, atropine, and suxam-ethonium regimen [5]. The SHINE trial is currently comparing the addition of a high-flow nasal cannula with standard care during neonatal intubation with a similar primary outcome (i.e., incidence of successful first attempt intubation without physiological instability) [6].

Successful neonatal intubation with the first attempt occurs in 64% of experienced professionals, but only in 20–26% of novice providers [2]. Thus, using continuous gas flow via the ETT might be an alternative technique especially for novice health care providers by which to reduce the risk for desaturation and/or bradycardia during intubation. As opportunities for endotracheal intubations of neonates are limited during pediatric training, the presented modified technique may lead to not only more stable infants, but also improved learning opportunities.

There are certain limitations to our study, which should be considered. The sample size was small and the study was stopped due to a change in the local intubation policy favoring less-invasive-surfactant-administration, which resulted in a lower intubation rate. We only report the p-values but do not mention significance, as this was a pilot trial with a small sample size and no power calculation. Hence, the results should be interpreted with caution. Furthermore, we only studied nasopharyngeal intubations, which represent the standard procedure at our unit. Using the same approach during oral intubation might yield different results. A strength of this study was the inclusion of small preterm infants, which are more prone to desaturation and/or bradycardia.

## 5. Conclusions

The concept of providing continuous gas flow through the ETT during nasopharyngeal intubation seems to be favorable without increased risk for major unexpected adverse events. This method might result in fewer intubation attempts and a higher rate of successful intubation on the first attempt and might reduce the number of abandoned intubations due to desaturation and/or bradycardia. Studies with greater sample sizes are urgently needed.

## Figures and Tables

**Figure 1 children-08-00553-f001:**
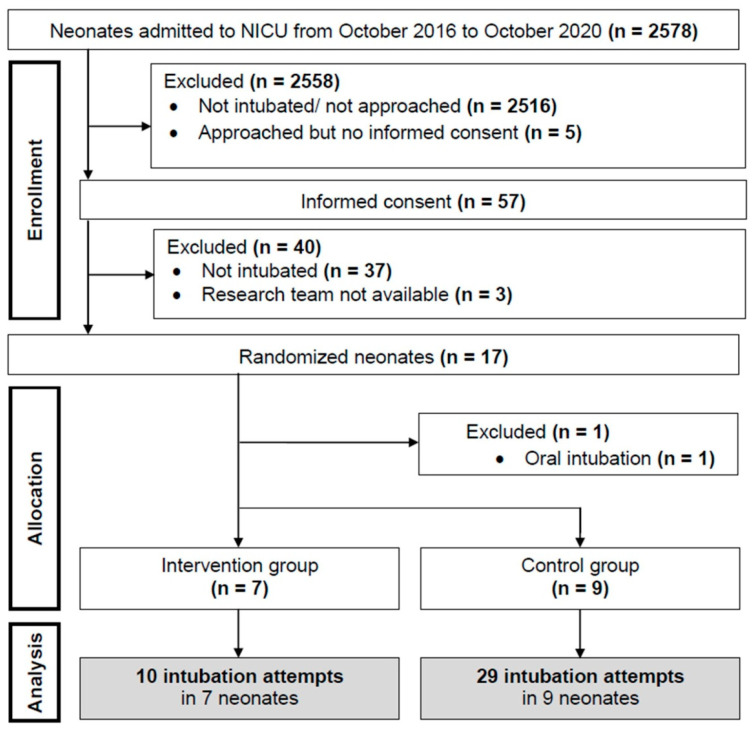
Flow Chart.

**Figure 2 children-08-00553-f002:**
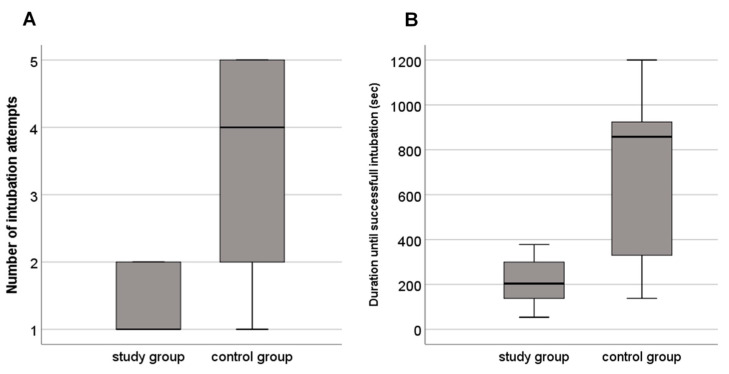
Boxplots of (**A**) number of intubation attempts and (**B**) duration until successful intubation in neonates intubated with the novel intubation method (study group) and the conventional intubation method (control group).

**Table 1 children-08-00553-t001:** Demographics and intubation characteristics. Data are expressed in *n* (%), median (IQR) or mean (SD) according to normal distribution. FiO_2_= Fraction of inspired oxygen.

	Study Group (*n* = 7)	Control Group (*n* = 9)	*p*-value
Demographics			
Gestational age (weeks)	30 (26–31)	32 (30–35)	0.222
Term infants (≥37 weeks gestation)	0 (0)	2 (22)	0.475
Female sex	2 (29)	5 (56)	0.608
Apgar 5 min	9 (8–9)	8 (8–9)	0.834
Postnatal age at intubation (hours)	15 (17)	36 (56)	0.873
Weight at intubation (grams)	1487 (819)	1857 (927)	0.461
Heart rate before intubation (beats/min)	151 (13)	145 (9)	0.292
Mean arterial blood pressure (mmHg) before intubation	44 (9)	42 (8)	0.694
Intubation			
Number of intubation attempts	1 (1–2)	4 (2–5)	0.056
Success on first intubation attempt	4 (57)	2 (22)	0.303
Duration until successful intubation (seconds)	204 (138–300)	858 (330–924)	0.114
Aborted intubations due to desaturation and/or bradycardia (n)	3	20	0.060
FiO2 during intubation	0.50 (0.40–0.75)	0.55 (0.40–0.70)	0.873
Gas flow (liters/minute)	6 (6–8)	8 (6–8)	0.289
Intravenous Propofol dosage (mg/kg)	1 (1–2)	2 (2–3)	0.072
Parameters of hospital stay			
Duration of invasive ventilation (days)	6 (2–7)	1 (0–3)	0.459
Duration of non-invasive ventilation (days)	14 (10–61)	6 (5–40)	0.153
Mortality (%)	0	0	1.000
Hospital stay (days)	46 (39–106)	30 (18–58)	0.187

## Data Availability

The data presented in this study are available on request from the corresponding author.

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
