# Peer review of "In Newborn Infants a New Intubation Method May Reduce the Number of Intubation Attempts: A Randomized Pilot Study"

_children, 2021, doi:10.3390/children8070553_

Round 1
Reviewer 1 Report
- The author presented a significant issue that deserve further studies. However, owing the small sample size, the statistical power of this study is not adequate. I suggested the author to enroll more cases and then complete this study.
- In addition, most of the subjects were preterm infants. Especially some patients in the intervention group is extremely preterm (GA < 28 weeks). Did these infants receive intubation in the delivery room? or in the NICU? In extremely preterm neonates, a lot of underlying diseases, like severity of respiratory distress syndrome will affect the intubation. The author seems not clarify this issue.
- The major problem of this study remains too small sample size.
Author Response
- The author presented a significant issue that deserve further studies. However, owing the small sample size, the statistical power of this study is not adequate. I suggested the author to enroll more cases and then complete this study.
Response: The authors understand your concerns and are aware of the small sample size. We stated this in results section (Page 3, Line 100) as well as in limitations (Page 5, Line 157) and mentioned in the discussion, that we only report the p-values but did not mention significance, as this was a pilot trial with small sample size and no power calculation.
Unfortunately, it is not possible to continue this study due to financial and organizational reasons. However, the authors are confident that the results of this study are valuable for neonatologists, since even in in this small sample size fewer intubation attempts, a shorter intubation duration and a reduced number of abandoned intubations were observed in the intervention group.
- In addition, most of the subjects were preterm infants. Especially some patients in the intervention group is extremely preterm (GA < 28 weeks). Did these infants receive intubation in the delivery room? or in the NICU? In extremely preterm neonates, a lot of underlying diseases, like severity of respiratory distress syndrome will affect the intubation. The author seems not clarify this issue.
Response: In our centre the NICU is located directly next to the delivery room, therefore all infants are resuscitated already at our NICU. The reviewer is right, this was not mentioned in this brief report. However, in Table 1 the postnatal age at intubation is demonstrated. This did not differ between the intervention and control group.
The reviewer is right that in extremely preterm neonates, a lot of underlying diseases, like severity of respiratory distress syndrome might affect the intubation. Indeed, the infants in the intervention group had a lower gestational age compared to those in the control group, which implicates that the new intubation method might be even more effective. We added this to the limitations and strength section (Page 5 Line 166-168).
- The major problem of this study remains too small sample size.
Response: Thank you for the valuable review. As mentioned above we are not able to continue this study and are confident that the results of this study are noteworthy and valuable for neonatologists.
Reviewer 2 Report
Thank you for this work. I found the manuscript quite interesting, addressing a practical, everyday issue for neonatologists. Although the sample size was very small, the results are noteworthy.
Neonates in the intervention group had longer hospital stay and duration on both invasive and non-invasive ventilation compared to the newborns of the control group. Could this possibly suggest a more severe respiratory status in the neonates in the control group? Taking into account the number of participants, might this have influenced the results? As expected, neonates with a more serious clinical course would present with more episodes of desaturation and bradycardia before intubation.
Please include in the abstract the number of the study participants. Also, the fact that the trial refers to nasopharyngeal intubation should also be mentioned.
And some minor suggestions:
Line 25-26: Modify to: “were 1 (1-2) versus 4 (2-5), (p=0.056) and n =3 versus n =20, (p=0.060) in the intervention and control group, respectively.”
Line 27: Change to “… fewer episodes of desaturation and/or bradycardia”
Line 43-44: Modify to “… flow during intubation (intervention group) will require less intubation attempts, compared to the standard approach without gas flow (control group)”.
Line 117: Change to “… no adverse effects in either group.”
Table 1: Please clarify the meaning of “FiO2 during to intubation”
Line 155-156: Modify to “may lead to not only more stable infants, but also improved learning opportunities.”
Line 160: Change to “do not mention significance”
Author Response
- Thank you for this work. I found the manuscript quite interesting, addressing a practical, everyday issue for neonatologists. Although the sample size was very small, the results are noteworthy.
Neonates in the intervention group had longer hospital stay and duration on both invasive and non-invasive ventilation compared to the newborns of the control group. Could this possibly suggest a more severe respiratory status in the neonates in the intervention group? Taking into account the number of participants, might this have influenced the results? As expected, neonates with a more serious clinical course would present with more episodes of desaturation and bradycardia before intubation.
Response: Thank you for this comment. The reviewer is right that hospital stay and duration of ventilation was longer in the intervention group. Furthermore, these infants had a lower gestational age compared to those in the control group (Table 1). These characteristics implicate that the infants in the intervention group might have suffered from more severe respiratory status compared to those in the control group. Therefore we suggest, that the new intubation method might be even more effective. We added this to the limitations and strength section (Page 5 Line 166-168).
- Please include in the abstract the number of the study participants. Also, the fact that the trial refers to nasopharyngeal intubation should also be mentioned.
Response: Thank you, correction done.
- And some minor suggestions: Line 25-26: Modify to: “were 1 (1-2) versus 4 (2-5), (p=0.056) and n =3 versus n =20, (p=0.060) in the intervention and control group, respectively.”
Response: Correction done.
- Line 27: Change to “… fewer episodes of desaturation and/or bradycardia”
Response: Correction done.
- Line 43-44: Modify to “… flow during intubation (intervention group) will require less intubation attempts, compared to the standard approach without gas flow (control group)”.
Response: Correction done.
- Line 117: Change to “… no adverse effects in either group.”
Response: Correction done.
- Table 1: Please clarify the meaning of “FiO2 during to intubation”
Response: We deleted the word “to”. The meaning of FiO2 is explained in the Tables legend.
- Line 155-156: Modify to “may lead to not only more stable infants, but also improved learning opportunities.”
Response: Correction done.
- Line 160: Change to “do not mention significance”
Response: Correction done. Thank you for the good and valuable review!
Round 2
Reviewer 1 Report
1. I still think this case number of this study was inadequate. I suggest to enroll more cases and re-do this study
Author Response
Dear Reviewer,
as mentioned in the revisions before it is not possible to continue this study due to financial and organizational reasons.
Kind regards,
Marlies Bruckner